# Co-Spray Dried Nafamostat Mesylate with Lecithin and Mannitol as Respirable Microparticles for Targeted Pulmonary Delivery: Pharmacokinetics and Lung Distribution in Rats

**DOI:** 10.3390/pharmaceutics13091519

**Published:** 2021-09-19

**Authors:** Ji-Hyun Kang, Young-Jin Kim, Min-Seok Yang, Dae Hwan Shin, Dong-Wook Kim, Il Yeong Park, Chun-Woong Park

**Affiliations:** 1College of Pharmacy, Chungbuk National University, Cheongju 28160, Korea; jhkanga@naver.com (J.-H.K.); 7777ytrewq@gmail.com (Y.-J.K.); pelomech@gmail.com (M.-S.Y.); dshin@chungbuk.ac.kr (D.H.S.); 2Department of Pharmaceutical Engineering, Cheongju University, Cheongju 28503, Korea; pharmengin@cju.ac.kr

**Keywords:** COVID-19, nafamostat mesylate, dry powder inhaler, lecithin, mannitol, co-spray

## Abstract

Coronavirus disease 2019 (COVID-19), caused by a new strain of coronavirus called severe acute respiratory syndrome coronavirus 2 (SARS-CoV-2), is spreading rapidly worldwide. Nafamostat mesylate (NFM) suppresses transmembrane serine protease 2 and SARS-CoV-2 S protein-mediated fusion. In this study, pharmacokinetics and lung distribution of NFM, administered via intravenous and intratracheal routes, were determined using high performance liquid chromatography analysis of blood plasma, lung lumen using bronchoalveolar lavage fluid, and lung tissue. Intratracheal administration had higher drug delivery and longer residual time in the lung lumen and tissue, which are the main sites of action, than intravenous administration. We confirmed the effect of lecithin as a stabilizer through an ex vivo stability test. Lecithin acts as an inhibitor of carboxylesterase and delays NFM decomposition. We prepared inhalable microparticles with NFM, lecithin, and mannitol via the co-spray method. The formulation prepared using an NFM:lecithin:mannitol ratio of 1:1:100 had a small particle size and excellent aerodynamic performance. Spray dried microparticles containing NFM, lecithin, and mannitol (1:1:100) had the longest residual time in the lung tissue. In conclusion, NFM-inhalable microparticles were prepared and confirmed to be delivered into the respiratory tract, such as lung lumen and lung tissue, through in vitro and in vivo evaluations.

## 1. Introduction

Coronavirus disease 2019 (COVID-19), which is caused by a new type of coronavirus called severe acute respiratory syndrome coronavirus 2 (SARS-CoV-2), is spreading rapidly worldwide. The first cases of COVID-19 outside of China were reported in January 2020, and the World Health Organization (WHO) declared the outbreak a pandemic. According to the World Health Organization (WHO 2020), approximately 70 million people have been infected with COVID-19 worldwide, and the death toll has reached 2.3 million in about 220 countries [1,2,3].

SARS-CoV-2 is an RNA virus that requires two steps to enter the human host. In the first step, the spike protein in the envelope of the virus is attached to a receptor on the cell surface, such as angiotensin-converting enzyme 2 (ACE2), which is then cleaved into S1 and S2 proteins by cellular proteases. Next, the S2 protein is cleaved by transmembrane serine protease 2 (TMPRSS2), cathepsin, or other proteases. This S2*′* cleavage is called priming, which aids the virus in fusing with the human cell membrane. These fusions lead to the formation of endosomes, aiding the viral RNA to penetrate the cell. The RNA of the virus that has penetrated the cell is replicated, and, eventually, the virus propagates. Therefore, suppression of the expression and function of ACE2 or TMPRSS2 can suppress SARS-CoV-2 infection and proliferation. In particular, previous studies have shown that TMPRSS2 knockout reduces the spread of SARS-CoV-2 and reduces lung damage in a mouse model [4,5,6,7,8,9].

Nafamostat mesylate (NFM) is a drug approved in South Korea and Japan for the treatment of acute pancreatitis and disseminated intravascular coagulation. It is a synthetic serine protease inhibitor that acts as an anticoagulant [3,10]. Recent studies have shown that NFM suppresses TMPRSS2 priming and inhibits MERS-COV membrane fusion [11]. It was also reported that NFM suppresses TMPRSS2 [12] and SARS-CoV-2 S protein-mediated fusion in the Calu-3 cell line—derived from epithelial cells of the lung—and reduces viral infection [5]. It has been reported that NFM inhibits SARS-CoV-2 entry into host cells at approximately 15-fold higher efficiency than camostat mesylate and at an EC50 in the low nanomolar range. In addition, it blocks SARS-CoV-2 infection in human lung cells with markedly higher efficiency than camostat mesylate [3]. 

However, NFM has a primary clinical drawback. It is easily eliminated by esterases in the blood and extracellular matrix, and, for this reason, its half-life in human blood is very short (approximately 3 min) [13,14,15]. Therefore, therapeutic doses of commercial NFM products are currently administered via continuous intravenous (IV) infusion [16]. Although these IV infusions may have an anticoagulation effect, they are inappropriate for treating and preventing SARS-CoV-2 because the pathway of SARS-CoV-2 penetration is via the respiratory tract [17,18]. When NFM is administered by IV infusion, the drug has difficulty reaching the respiratory tract because the systemic circulation system is esterase-rich. In particular, NFM is inefficient in controlling the invasion and growth of viruses because it is almost impossible for the drugs to reach the respiratory niches. In addition, when high doses of the drug are administered by IV infusion to achieve therapeutic blood concentrations, anticoagulant side effects are experienced by the patient. Therefore, to treat and prevent SARS-CoV-2 using NFM, it is necessary to administer the drug via the respiratory tract, which is one of the routes of viral invasion and has a lower esterase level than the blood [19]. When the drug is administered via the respiratory tract, it results in high efficacy and reduced incidence of side effects [20,21,22,23]. 

Dry powder inhalation (DPI) is a pulmonary delivery system that utilizes the breathing process to disperse and deliver micronized solid drug particles to the lungs. It is preferred to liquid-based systems such as nebulizers and pressurized metered dose inhalers (pMDIs). DPI is portable, propellant-free, easy to operate, has a flexible-dose capacity, and because of its dry state, it is highly stable [24]. The aerosol performance of the particles is influenced by physicochemical properties such as particle size distribution, particle shape, surface roughness, interparticle force, and solid-state, which determine the efficiency of the DPI. In addition, it should be mentioned that microparticles with an aerodynamic size of 1–5 μm are required for pulmonary delivery [25,26,27].

Lecithin is widely used in the pharmaceutical industry as a dispersing, emulsifying, and stabilizing agent [28]. Lecithins are used in formulations meant for intravenous, intramuscular, topical, oral, rectal, and pulmonary administration. Lecithin has been widely investigated for its role in enhancing drug bioavailability. It enhances drug stability and absorption to improve bioavailability when added to microemulsions or solid lipid nanoparticles [29,30,31]. Moreover, lecithin acts as an inhibitor of carboxylesterase and can delay the decomposition of drugs by esterases. [32,33,34]. Since NFM is rapidly decomposed by esterases in the human body and blood, lecithin is expected to improve the stability of NFM and enhance its bioavailability. In addition, the half-life of NFM may also be increased when it is administered with ester group-rich substances such as lecithin, which may suppress NFM degradation by respiratory lumen esterases. [13,35,36].

Here, we conducted a pharmacokinetic and lung distribution study of NFM, administered via intravenous and intratracheal routes, by assessing its concentration in the blood, bronchoalveolar lavage fluid (BALF), and lung tissue. Second, we determined the effect of substances rich in ester groups such as lecithin [33,37,38], which can act as inhibitors of esterases, to delay the degradation of NFM and consequently improve NFM chemical stability in the body. Finally, we prepared a DPI formulation of NFM as an inhalable microparticle of NFM, mannitol, and lecithin using the co-spray method. We hypothesized that this formulation would be effective in increasing the NFM elimination half-life after intratracheal administration for COVID-19 treatment through in vitro and in vivo evaluations.

## 2. Materials and Methods

### 2.1. Materials

NFM was purchased from Kukjeon Pharmaceutical (>99% purity, JP grade, Anyang, Korea). Soybean lecithin (Extra pure) and acetic acid were purchased from Samchun Chemical (Pyeongtaek, Korea). D-mannitol was obtained from Whawon Pharmaceutical (Hwaseong, Korea). Sodium 1-heptane sulfonate was purchased from Tokyo Chemical Industry (Tokyo, Japan). High-performance liquid chromatography (HPLC)-grade ethanol and acetonitrile were purchased from Honeywell Burdick and Jackson (Muskegon, MI, USA). Water was purified by filtration in the laboratory. HPLC-grade solvents were used for analyses. Sprague–Dawley (S.D.) rats were purchased from the Samtako Corporation (Osan, Korea).

### 2.2. Methods

#### 2.2.1. Preparation of NFM Microparticles Using the Co-Spray Dry Method

To prepare the NFM microparticles, the co-spray-drying process was performed using a laboratory-scale spray dryer (EYELA SD-1000, Rikakikai, Tokyo, Japan). The feeding solution was prepared by completely dissolving NFM, lecithin, and mannitol in 80% ethanol (*v*/*v*) at ratios of 1:1:100, 1:1:50, and 1:1:10 (*w*/*w*/*w*) to obtain a total powder concentration of 2% (*w*/*v*). The following parameters were used during spray-drying: inlet temperature, 100 °C; outlet temperature, 45–48 °C; nozzle size, 0.4 mm, feed rate, 10 mL min^−1^; atomization air pressure, 200 kPa; and drying air flow rate, 0.30 m^3^ min^−1^. The spray dried NFM, lecithin, and mannitol (SD-NLM) were SD-NLM1: 1:1:100, SD-NLM2: 1:1:50, and SD-NLM3: 1:1:10, depending on the ratio of NFM:lecithin:mannitol. The NFM microparticles were kept in a glass vial containing silica gel at 20 °C, 40% relative humidity (RH) until use.

#### 2.2.2. Physicochemical Characterization of NFM Microparticle

SD-NLMs were characterized according to their morphology and size distribution. The morphology was examined using scanning electron microscopy (SEM) (ZEISS-GEMINI LEO 1530, Zeiss, Germany). The raw NFM and spray dried NFM (SD-NFM) microparticles were spread on a carbon tape and the unattached microparticles were blown off; the NFM microparticles were then platinum coated to a thickness of 200 Å using a Hummer VI sputtering device (Anatech, Sparks, NV, USA). Magnifications of ×1000 and ×3000 and a voltage of 3 kV were used.

The particle size distribution of the raw NFM and SD-NFM microparticles was determined using laser diffraction particle sizing (Mastersizer3000, Malvern Instruments, Malvern, UK) by the dry dispersion method. The scattering model uses the Mie scattering theory. Each measurement was conducted in triplicate, and the means and standard deviations were calculated.

In addition, X-ray diffraction (XRD) patterns of the NFM raw material, excipients, physical mixture, and SD-NLMs were analyzed using a D8 Discover with a general area detector diffraction system (GADDS) (Bruker AXS, Karlsruhe, Germany) at a wavelength of 1.54 Å. The 2θ scans were conducted between 5° and 60°. 

Infrared spectroscopy was performed using a Fourier-transform infrared spectroscopy (FT-IR) spectrophotometer (4100 Jasco, Tokyo, Japan). For each spectrum, 16 transient spectra were collected over the range 650–4000 cm^−1^ for the NFM, lecithin, mannitol, the physical mixture, and the prepared SD-NLMs. 

The thermal properties of the NFM raw material, other excipients, and prepared SD-NLMs were analyzed using a DSC Q2000 (TA Instruments, New Castle, DE, USA) thermal analyzer system. The samples were accurately weighed, loaded into an aluminum pan, and analyzed at a heating rate of 10 °C min^−1^ over a temperature range of 30–300 °C. The thermal response of the prepared sample was calculated using the TA Advantage/Universal Analysis software (v5.2.6, TA Instruments, New Castle, DE, USA). 

#### 2.2.3. In Vitro Aerodynamic Performance Evaluation of NFM Microparticles

Based on the United States Pharmacopeia (USP) Chapter 601 specification of aerosols, the aerosol performance of the SD-NLMs DPI was determined using an 8-stage non-viable ACI (TE-20-800, TISCH Environmental, Cleves, OH, USA) and RS01*^®^* (Plastiape, Osnago, Italy). To prevent particle bouncing and re-entrainment, the collection plates of the ACI stage were pre-coated with silicone oil [20]. SD-NLMs equivalent to 10 mg of formulation were loaded into a hydroxypropyl methylcellulose hard capsule (HPMC, size 3). Each formulation was aerosolized in an amount equivalent to 1 mg of NFM for each experiment, with air drawn through at a controlled flow rate of 28.3 L·min^−1^ for 4 s. The quantity of NFM microparticles remaining in the capsule and deposited onto each collection plate of the stage was measured using HPLC [39]. The HPLC system (Ultimate 3000 series HPLC system, Thermo Scientific, Waltham, MA, USA) was operated at 260 nm with a Luna L11 150 mm × 4.60 mm, 5 μm column (Phenomenex, Torrance, CA, USA). The mobile phase consisting of acetonitrile and buffer (0.1 M acetic acid and 0.03 M sodium 1-heptane sulfonate) in a 30:70 (*v*/*v*) ratio was eluted at a flow rate of 1.0 mL min^−1^. The column temperature was maintained at 40 °C, and the volume of each injected sample was 20 μL. The HPLC method was validated (Appendix A). The emitted dose (ED) and fine particle fraction (FPF) were calculated using the following Equations:Emitted dose [ED, %] = [Initial mass in capsule-final mass remaining in the capsule]/[Initial mass in capsule](1)
Fine particle fraction [FPF, %] = [Mass of the particle in stages 2 through 7]/[Mass of the particle in all stages](2)

The mass median aerodynamic diameter (MMAD) and geometric standard deviation (GSD) were calculated using information from USP Chapter 601. MMAD was determined from a plot of the percentage of mass lower than the stated aerodynamic diameter versus the aerodynamic diameter, D_50%_, on a log probability paper. The GSD was calculated using the following Equation: Geometric standard deviation [GSD] = √[D_84.13%_/D_15.87%_](3)

#### 2.2.4. In Vitro Dissolution Test of NFM Microparticles

The in vitro dissolution behavior of the spray dried NFM and mannitol (1:100 *w*/*w*, SD-NM) was evaluated using a Franz diffusion cell system (FCDS-900C, Labfine Instruments, Anyang, Korea). The receptor compartment of the Franz diffusion cell was filled with 12 mL of distilled water (DW), which was maintained under sink conditions for the experiment [40,41]. A cellulose membrane filter (pore size, 0.45 μm, ADVANTEC, Tokyo, Japan) was used as a barrier and was placed on the receptor. The membrane was in contact with the receptor medium, which led to the formation of an air–liquid interface [42]. The receptor medium was maintained at 37 ± 1 °C and continuously stirred to ensure homogeneity. SD-NM and SD-NLM equivalent to 1 mg of NFM were uniformly spread on the surface of the membrane at the air–liquid interface. At a defined time, 200 μL of the medium were added, and the same volume of fresh DW was added. The content of the drug was quantified using HPLC, as described above.

#### 2.2.5. Ex-Vivo Stability Test of NFM in BALF

We prepared solution 1 (S1) at a concentration of 200 μg·mL^−1^ of NFM in DW. Lecithin was added to S1. to prepare lecithin concentrations of 100 μg·mL^−1^, 200 μg·mL^−1^, and 400 μg·mL^−1^, which were labeled as solution 2 (S2), solution 3 (S3), and solution 4 (S4), respectively. BALF was obtained by injecting and extracting 5 mL of DW into the lungs of Sprague–Dawley rats (SD rats, male, 8 weeks old) [43,44]. S1, S2, S3, and S4 were diluted 10-fold with BALF to prepare 20 μg·mL^−1^ NFM as samples. Each sample was stored in an oven at 37 °C. The samples were quantified through HPLC analysis immediately after preparation (initial) and on the first, second, and third days after storage. Before HPLC analysis, acetonitrile (ACN) was added to each sample at a ratio of 1:1 (*v*/*v*) to precipitate the proteins, and the samples were centrifuged at 18,000 rpm for 5 min to obtain a supernatant. The supernatant was quantified by HPLC analysis.

#### 2.2.6. In Vivo Pharmacokinetics and Efficacy Study of NFM 

The Chungbuk National University Institutional Animal Care and Use Committee approved the experimental protocols and animal care methods used in this study. The first in vivo experiment was a pharmacokinetic study to determine the efficacy of NFM based on the route of administration and to determine the effect of lecithin as a stabilizer. After preparing a 0.5% mannitol solution (*w*/*v*), NFM was added at a concentration of 25 mg mL^−1^ to prepare solution A. In addition, Tween 80 was added to solution A at 1% (*w*/*v*), plus lecithin, which was added at a concentration of 25 mg mL^−1^, and stirred to evenly disperse it to produce solution B. Each experiment was carried out using a male SD rat (8 weeks old, 230–250 g) in three groups of 16 animals each. The SD rats were fed a commercial pellet diet and fresh water and housed at room temperature (23 ± 1 °C) with a relative humidity of 50 ± 10%, and a 12 h light and dark cycle. First, the intravenous (IV) group was administered a 100 μL dose of solution A (10 mg·kg*^−^*^1^) intravenously through the lateral tail vein. Second, a 100 μL dose of solution A (10 mg kg*^−^*^1^) was administered by intratracheal instillation (ITI) to the intratracheal instillation-1 (ITI-1) group. Finally, the ITI-2 group was administered a 100 μL dose of solution B (10 mg·kg*^−^*^1^). For the analysis of plasma concentration, blood was collected via the orbital vein 1, 5, 10, 30, and 60 min after administration to each group (*n* = 4) [45]. Blood was collected in a heparin tube and gently shaken. Plasma was immediately separated from the blood samples by centrifugation at 3000 rpm for 2 min. After administration to each group (*n* = 12), 4 rats were sacrificed at 10, 30, and 60 min. BALF samples were obtained by injecting and extracting 5 mL DW into the lungs of the rats. Lung tissues were placed in 10 mL DW and homogenized at 18,000 rpm for 3 min using a homogenizer. Acetonitrile (ACN) was added to plasma, BALF, and lung tissue samples in a ratio of 1:1 (*v*/*v*) to precipitate the proteins, and the sample solutions were centrifuged at 18,000 rpm for 5 min to obtain a supernatant. The supernatant was quantified by HPLC analysis. A second in vivo experiment was performed to determine the efficacy of NFM microparticles in a DPI formulation, focusing on the lung lumen and tissue. First, the solution group was administered a solution at a dose of 0.25 mg NFM per rat by ITI (1 mg·kg*^−^*^1^). Second, the SD-NM group was treated using a dry powder insufflator for the intratracheal administration of the SD-NM in the form of DPI at a dose of 0.25 mg of NFM per rat (1 mg·kg*^−^*^1^). Finally, the SD-NLM group was administered an SD-NLM at a dose of 0.25 mg of NFM per rat (1 mg kg*^−^*^1^) using a dry powder insufflator. BALF and lung tissue samples were prepared using the same procedure as that used for the in vivo study described above. The samples were quantified by HPLC analysis.

### 2.3. Statistical Analysis 

Statistically significant differences were evaluated using a one-way analysis of variance (ANOVA) with least significant difference (LSD) or Games–Howell post hoc tests and Student’s t-test using SPSS version 23 (SPSS, Chicago, IL, USA). Statistical significance was set at *p* < 0.05.

## 3. Results and Discussion

### 3.1. Preparation of NFM Microparticles by the Co-Spray Dry Method

Spray dried microparticles containing NFM, lecithin, and mannitol (SD-NLM) were prepared using the co-spray dry method. We used NFM, which is the active pharmaceutical ingredient, lecithin as an NFM stabilizer, and mannitol, which is widely used as a diluent agent in DPIs. The ratio of NFM to lecithin was fixed at 1:1 in the stability test in the BALF, Figure 5, and the ratio of mannitol was varied at 1:1:0, 1:1:10, 1:1:50 to 1:1:100 (*w*/*w*). We found that the higher the proportion of mannitol, the higher the yield, and the same was found for the smaller particle sizes and lower span values (Table 1). The yields were 42.1, 44.7, 21.5, and 5.1% for SD-NM, SD-NLM1, SD-NLM2, and SD-NLM3, respectively. Dv50 was 2.40 ± 0.021, 2.60 ± 0.010, 2.58 ± 0.015 and 10.27 ± 0.153 μm (mean ± SD), respectively. As the proportion of liquid lecithin increased, the yield decreased. In addition, Because of the viscosity of lecithin, particles formed clusters and existed in large particles. These results can be confirmed by comparing SD-NM and SD-NLM1, their Dv10 values were 0.27 ± 0.002 and 1.36 ± 0.000 μm, respectively, which was about five times different depending on the presence or absence of lecithin. Considering the particle size and yield, the optimal formulation was SD-NML1, and the optimal NFM: lecithin: mannitol ratio was 1:1:100.

### 3.2. Physicochemical Characteristics of NFM Microparticles

The physicochemical characterization of NFM microparticles was performed. First, the morphological characteristics were evaluated using SEM Figure 1. Raw NFM had an irregular shape and wide particle sizes ranging from tens to hundreds of microns. Raw mannitol had a cubic shape and a larger particle size than raw NFM. In contrast, SD-NM and SD-NLM1 had a spherical shape with a smooth surface and smaller particle size than the raw materials. SD-NLM2 and SD-NLM3 appeared as small particle clusters, which are thought to be due to the high proportion of lecithin acting as a binder. In conclusion, the morphological evaluation using SEM showed similarity with the particle size distribution shown in Table 2.

Differential scanning calorimetry (DSC) thermograms of the raw materials and spray dried NFM microparticles are shown in Figure 2A. Melting peak of NFM was observed at 263 °C. However, the spray dried NFM (SD-NFM) seems to have lost crystallinity as no melting peak was found. Lecithin was subjected to DSC analysis at temperatures up to 270 °C, no melting occurred, but a small and broad glass transition was observed at 195 °C. The thermogram of mannitol showed a sharp endothermic peak at 166 °C. This is similar to the previously reported T_m_ of mannitol. Most of the thermograms of the spray dried NFM microparticles were affected by mannitol. In addition, as the T_m_ peak of NFM was very broad in SD-NLMs, NFM present inside SD-NLMs is considered to be in partial crystal form. SD-NM showed an endothermic peak at 166 °C, which was thought to be due to mannitol. This means that when the co-spray was dried with mannitol and NFM, there was no interaction between the two components, and the crystals of mannitol did not change significantly. SD-NLM1, 2, and 3 had T_m_ values of 165.5, 164, and 163.5 °C, respectively. The higher the proportion of lecithin, the lower the T_m_. However, this difference was not statistically significant. X-ray diffraction (XRD) was used to evaluate the crystallographic properties of the spray dried NFM microparticles. The XRD data shown in Figure 2B corroborate the DSC results. XRD analysis offers the advantage of allowing the identification of the crystalline or amorphous state of microparticles and their crystal modification. Raw NFM exhibited sharp and narrow diffraction peaks at 2θ = 12°, 16°, 24°, and 29°, which are characteristic of a high degree of long-range molecular order. Moreover, raw mannitol showed peaks at 2θ of 10°, 14°, and 44°, and the mannitol peaks contained in the spray dried NFM microparticles were low. Furthermore, the diffraction pattern of the prepared NFM microparticles differed considerably from that of the raw NFM. The peaks of raw NFM mostly disappeared in the XRD pattern of NFM microparticles, indicating that the crystal arrangement of NFM in microparticles was less ordered than that in raw NFM. The loss of crystallinity in the XRD pattern of the spray dried NFM indicated the absence of sharp peaks and the loss of most of the peaks of raw NFM. Through comparison with earlier results, it was concluded that the drug molecularly dispersed in the prepared NFM microparticles and was in an amorphous or disordered crystalline phase [46]. For the final physicochemical characterization, FT-IR spectra are shown in Figure 2C. The spectrum of raw NFM showed strong infrared absorption at 1740–1680 cm^−1^. These peaks correspond to C–H bonds [20], and broad infrared absorption at 3440–3000 cm^−1^ corresponds to N-H bonds [47] of guanidine and amidine groups. In the case of SD-NFM, although it shows a peak similar to the main peaks of raw NFM, the peak is weakened and broadened. This is thought to be because the crystalline form has changed to an amorphous form. Lecithin showed peaks at 2940 cm^−1^ and 1730–1680 cm^−1^, which are attributed to C-H bonds, with the 2940 cm^−1^ peak corresponding to the NFM microparticles. In addition, SD-NLM3, which has the highest proportion of lecithin, also showed a weak peak at 1730–1680 cm^−1^. Mannitol, similar to NFM, showed broad infrared absorption at 3400–3000 cm^−1^, which is thought to result from the O-H bond. The NFM microparticles also showed absorption at 3400–3000 cm^−1^. Considering the wave number and shape of the peak, this peak corresponds to mannitol rather than NFM. In conclusion, through the study of physicochemical properties such as DSC, XRD, and FT-IR, it is hypothesized that NFM has changed crystallinity or is amorphous in spray dried microparticles.

### 3.3. In Vitro Aerodynamic Performance Evaluation of NFM Microparticle

The aerosol dispersion performance of the spray dried microparticles was evaluated using an Andersen cascade impactor (ACI) and RS01 as a dry powder inhalation device. Figure 3 presents the distribution of NFM as the percent deposition and deposited amount of each stage. SD-NLM2 and SD-NLM3 with large particle sizes were deposited by 50% or more in 0 stages, and the amount of deposit was reduced as the cut-off diameter was smaller. In contrast, SD-NM and SD-NLM1 with small particle size and span values had a large amount of deposits in the 2–7 stage with a cut-off diameter of less than 5 μm. In particular, SD-NM with the smallest Dv10 was distributed up to the stage with a smaller cut-off than SD-NLM1. 

The mass median aerodynamic diameter (MMAD), geometric standard deviation (GSD), emitted dose (ED), and fine particle fraction (FPF) values obtained by the ACI deposition study are listed in Table 2. The ED was the lowest for SD-NM, which had the smallest particles [30], but showed acceptable results of over 90%. The FPF, which indicates the rate of reaching the alveolar region, was 75.1% and 50.8% for SD-NM and SD-NLM1, respectively; these values were significantly higher than those of the other two forms (*p* < 0.005). The total deposited amount was the highest in SD-NLM1 (74.6%), which was significantly higher than that of SD-NLM3. Although not analyzed, the unknown amount is presumed to be buried in the induction port or device. The MMADs of SD-NM and SD-NLM1 were 2.1 μm and 3.5 μm, respectively, whereas the GSD values were 2.6 and 3.0, respectively, which were significantly smaller than those of the other two formations (*p* < 0.005). As expected, SD-NM and SD-NLM1 showed better results than the other formulations, which differed significantly in aerodynamic performance owing to their smaller particle sizes. SD-NM showed higher FPF and lower MMAD than SD-NLM1 but also had a tolerable value of FPF for aerodynamic performance. In conclusion, among the NFM microparticles obtained by using the co-spray dry method, the SD-NLM1 formulation prepared with an NFM:mannitol ratio of 1:100 had a small particle size and excellent aerodynamic performance, thus the optimum ratio was 1:100. In addition, these SD-NM and SD-NLM1 formulations have excellent DPI characteristics. In particular, the MMAD of SD-NM and SD-NLM1 ranged from 1 to 3 μm, referring to the respirable particle size delivering optimally to the respiratory bronchioles and alveolar region, which are target sites of inhaled NFM for the treatment of COVID-19 [20,42,48,49]. 

### 3.4. In Vitro Dissolution Test of NFM Microparticles

Figure 4 shows the dissolution profiles of SD-NM and SD-NLM1 obtained using a Franz diffusion cell. SD-NM showed a significantly higher release rate than SD-NLM1, which was approximately two times higher at 60 min, but both formulations released less than 10% in 1 h (*p* < 0.05, ANOVA). This is thought to be because SD-NLM1 contains lecithin, which increases the hydrophobic properties of the microparticles and makes wetting difficult, thereby slowing the drug release [50,51]. It is known that esterases are the reason NFM is rapidly degraded when administered [13,35,36]. For interaction with esterase, NFM should not be present in the microparticle and must be released and dissolved in the molecular state. Therefore, lecithin, which has the effect of sustaining the release of NFM, can improve the sustainability of NFM in the lung lumen.

In the in vitro stability test in BALF and in vivo pharmacokinetic studies of the NFM solution (Section 3.5), lecithin improved the stability of NFM. In addition, lecithin may have acted as an esterase inhibitor, and, from the results of the dissolution test, it suppressed the release of NFM from microparticles. These two properties of lecithin would, therefore, improve the stability and sustainability of NFM in the lung lumen. Consequently, SD-NLM1 containing lecithin is expected to be more stable and long-acting than SD-NM when administered as a DPI.

### 3.5. In Vitro Stability Test of NFM in BALF

The mechanism by which NFM acts as a therapeutic agent for COVID-19 is by inhibiting the penetration of SARS-CoV-2. Therefore, the main target site of NFM is the lung lumen, which has a very large surface area. We conducted an in vitro stability test of NFM in the BALF to confirm whether NFM remains stable and effective when administered by inhalation (Figure 5). In addition, lecithin had an inhibitory effect on the degradation of NFM, and we, therefore, optimized the ratio of NFM to lecithin. Consequently, S1 was the only NFM solution in the BALF, and almost all NFMs in S1 were decomposed in a day. In contrast, the S2, S3, and S4 groups with NFM: lecithin ratios of 2:1, 1:1, and 1:2, respectively, were found to be more stable than the S1 group (*p* < 0.005). It was observed that the higher the proportion of lecithin, the higher the residual amount of NFM, and the more stable it was. In conclusion, NFM was rapidly degraded in the blood and the BALF (or lung lumen) due to the presence of esterase in these locations. We found that lecithin stabilized NFM to enhance its activity. We observed that the lecithin: NFM ratio that stabilizes NFM was 1:1 and proceeded to develop a DPI for NFM in this ratio.

### 3.6. In Vivo Pharmacokinetics Study of NFM Solution

The plasma concentration of NFM was studied after either intravenous (IV) or intratracheal instillation (ITI) administration of NFM (Figure 6). There were two ITI groups: ITI-1 (without lecithin) and ITI-2 (with lecithin). The blood concentration of the IV group was drastically reduced 1 min after the initial sampling. However, the ITI-1 and ITI-2 groups reached C_max_ 5 min after administration and plasma concentration of NFM gradually decreased because it took a while for the drug to be absorbed and transported from the lungs to the blood. 

The pharmacokinetic parameters are shown in Table 3. The area under the curve (AUC) was 436, 675, and 730 min·μg mL^−1^ for the IV, ITI-1, and ITI-2 groups, respectively. The ITI-1 group had higher AUC than the IV group (*p* < 0.05). The ITI-2 group also had a significantly higher AUC than the IV group (*p* < 0.005). Similarly, the half-life of the ITI-1 and ITI-2 groups was approximately twice that of the IV group (*p* < 0.005). NFM was degraded faster in the IV group because the blood is richer in esterases than the lung. It is also believed that the NFM in the ITI groups was gradually absorbed from the lungs into the blood, thus maintaining plasma concentration and increasing the AUC and half-life. 

The amount of NFM in the BALF over time is shown in Figure 7A. The amount of drug in the BALF was considered the amount of drug in the lung lumen, which is a site where the action of NFM, i.e., substantial suppression of SARS-COV-2 penetration, is required. The amount of NFM in the BALF 10 min after administration was 0.96 μg·mL^−1^, 2.90 μg·mL^−1^, and 4.48 μg·mL^−1^ for the IV, ITI-1, and ITI-2 groups, respectively (*p* < 0.05). There was no NFM in the BALF of the IV group after 30 min. In contrast, almost the same amount of NFM remained in the BALF after 30 min, i.e.*,* 1.69 and 1.69 μg·mL^−1^, in the ITI-1 and ITI-2 groups, respectively. After 60 min, only the ITI-2 group still had remaining NFM in the BALF. Figure 7B shows the amount of drug in the lung tissue. Some drug remained in the lung tissue of the IV group 10 min after administration, but none remained afterward. The IV group also showed a significantly lower amount of NFM in the lung tissue than the other groups (*p* < 0.005). The ITI-1 and ITI-2 groups showed similar amounts of NFM after 10 min. However, after 30 min, the concentration of NFM in the lung tissue of the ITI-1 and IT1-2 groups was 15.23 μg·g^−1^ and 44.19 μg·g^−1^, respectively (*p* < 0.005). After 60 min, only the ITI-2 group had some residual drug in the lung tissue, as was the case in the BALF. Based on the results of the two studies, the ITI-1 group delivered more drug than the IV group and the drug remained in the lung lumen and lung tissue for a longer period for the former. These findings mean that NFM, which suppresses the penetration of SARS-CoV-2, is more efficiently delivered to the lung lumen when administered via the inhalation route as compared with the IV route. This also means that the inhalation route is more effective than the IV route for the retention of NFM in the lungs. Furthermore, by comparing the ITI-1 and ITI-2 groups to determine the effect of lecithin, it was found that more NFM from ITI-2 administration remained in the lung tissue than that from ITI-1 administration after 30 min. After 60 min, there was some drug remaining in the BALF and lung tissue only in the ITI-2 group. These results indicate that lecithin is an effective inhibitor of NFM degradation. In conclusion, NFM is a highly efficient therapeutic agent for COVID-19 when administered via the inhalation route. This route leads to higher drug delivery and longer residual time in the lung lumen and lung tissue, which are the main sites of action, than the IV route. In addition, when NFM is administered with lecithin, it may suppress the degradation of NFM, thereby increasing the action duration.

### 3.7. In Vivo Pharmacokinetics Study of NFM Microparticles

An in vivo efficacy study was performed after the administration of NFM microparticles prepared using the co-spray dry method. NFM solution, SD-NM, and SD-NLM1 were each administered at a dose of 0.25 mg NFM per rat (1 mg kg*^−^*^1^). SD-NM and SD-NLM1 were administered as DPIs using a dry powder insufflator (DP-4, Penn-Century, Philadelphia, PA, USA). The solution was administered by the ITI method. Figure 8A shows the amount of NFM in the BALF after administration, indicating the amount of NFM in the lung lumen. Ten minutes after administration, there was more NFM after administration of SD-NM (2.1 μg·mL^−1^) and SD-NLM1 (2.3 μg·mL^−1^) than of the solution (0.9 μg·mL^−1^) (*p* < 0.05). Thirty minutes later, the solution, SD-NM, and SD-NLM1 remained at 0.3 μg·mL^−1^, 0.7 μg·mL^−1^, and 1.2 μg·mL^−1^, respectively. Therefore, more NFM remained in the BALF after the administration of SD-NM compared to NFM from solution (*p* < 0.05). Furthermore, in the case of SD-NLM1 administration, more NFM also remained in the tissue than by treatment with SD-NM (*p* < 0.05) or solution (*p* < 0.005). After 60 min, the solution group had no NFM left in the BALF and more NFM for SD-NLM1 than for SD-NM administration (*p* < 0.05). It is expected that the solution is rapidly absorbed and distributed to the lung tissue and blood or is rapidly metabolized by esterases. SD-NLM1 contains lecithin, and based on our dissolution results, this delayed the release of NFM from microparticles. Because of the delayed release of NFM, the absorption, distribution, and metabolism of NFM were delayed, and the amount of drug in the BALF was maintained longer than for SD-NM. In addition, we found that lecithin is an esterase inhibitor, and, therefore, protects NFM from degradation. The amount of drug remaining in the lung tissue is shown in Figure 8B. After 10 min, 5.7 μg·g^−1^, 2.0 μg·g^−1^, and 2.0 μg·g^−1^ of NFM remained in the lung tissue for the solution, SD-NM, and SD-NLM1 groups, respectively. There was no significant difference between the three groups at 30 min. After 60 min, the amounts of drug remaining in the lung tissue for the solution, SD-NM and SD-NLM1 groups were 0.4 μg·g^−1^, 0.7 μg·g^−1^, and 1.1 μg·g^−1^, respectively. This shows that the amount of drug retained in the lung tissue of the SD-NLM1 group was approximately three times that of the solution group and approximately twice that of the SD-NM group (*p* < 0.05). The results of the in vitro dissolution test showed that NFM from SD-NM was released faster than from SD-NLM1. The faster the drug was released, the faster the absorption and the higher the amount of NFM in the lung tissue, similar to that of the solution group. However, SD-NM released an equal or smaller amount of drug in the lung tissue than SD-NLM1. The amount of drug in the BALF and lung tissue showed that lecithin reduced the release rate of NFM and inhibited its degradation by esterases, and, therefore, SD-NLM1 is the form that leads to longest NFM retainment in the lung tissue. In conclusion, we prepared co-spray dried NFM microparticles with lecithin and mannitol, which showed efficient drug delivery into the respiratory tract through in vitro and in vivo evaluations.

## 4. Conclusions

In this study, we conducted a pharmacokinetic and efficacy study based on the route of administration of NFM, which is known to have a therapeutic effect on COVID-19. We found that administration via the inhalation route was more advantageous than IV injection in the sustainability of NFM, and the residual amount and residual time were higher in the lung lumen and lung tissue, which are the main target sites of NFM. In addition, we found that lecithin could act as an NFM stabilizer as an esterase inhibitor. After studies to prove these concepts, inhalable NFM microparticles were prepared via the co-spray dry method using NFM, lecithin, and mannitol. Through an in vivo study, it was confirmed that lecithin-containing NFM microparticles (SD-NLM1) provide more efficient drug delivery into the lung lumen and tissue. Therefore, the formulated inhalable NFM microparticles can be considered as a potential pulmonary drug delivery system for the treatment of COVID-19.

## Figures and Tables

**Figure 1 pharmaceutics-13-01519-f001:**
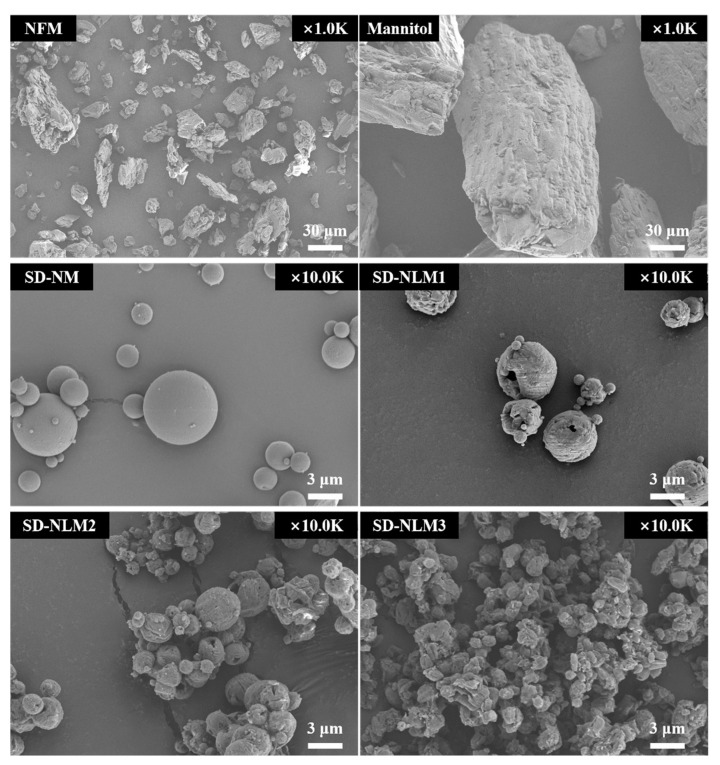
Scanning electron microscopy (SEM) image of nafamostat mesylate (NFM) microparticles. SD-NM: spray dried NFM and mannitol (1:100); SD-NLM1: spray dried NFM, lecithin, and mannitol (1:1:100); SD-NLM2: spray dried NFM, lecithin, and mannitol (1:1:50); SD-NLM3: spray dried NFM, lecithin, and mannitol (1:1:10).

**Figure 2 pharmaceutics-13-01519-f002:**
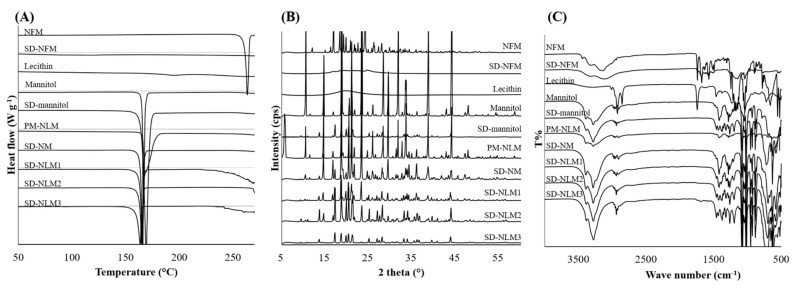
Physicochemical properties of nafamostat mesylate (NFM) microparticles. (**A**) Differential scanning calorimetry (DSC) thermograms of raw materials and prepared NFM microparticles. (**B**) X-ray diffraction (XRD) patterns of the raw materials and prepared NFM microparticles. (**C**) Fourier-transform infrared (FT-IR) spectra of the raw materials and prepared NFM microparticles. SD-NFM: spray dried NFM; SD-NM: spray dried NFM and mannitol (1:100); SD-NLM1: spray dried NFM, lecithin and mannitol (1:1:100); SD-NLM2: spray dried NFM, lecithin and mannitol (1:1:50); SD-NLM3: spray dried NFM, lecithin and mannitol (1:1:10).

**Figure 3 pharmaceutics-13-01519-f003:**
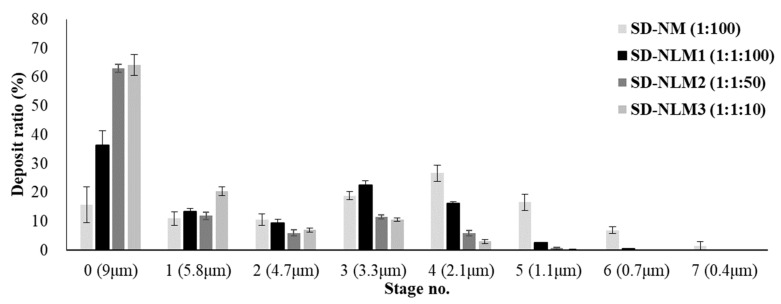
In vitro aerosol performance in each stage of Andersen cascade impactor (mean ± standard deviation, *n* = 3). SD-NM: spray dried NFM and mannitol (1:100); SD-NLM1: spray dried NFM, lecithin and mannitol (1:1:100); SD-NLM2: spray dried NFM, lecithin and mannitol (1:1:50); SD-NLM3: spray dried NFM, lecithin and mannitol (1:1:10).

**Figure 4 pharmaceutics-13-01519-f004:**
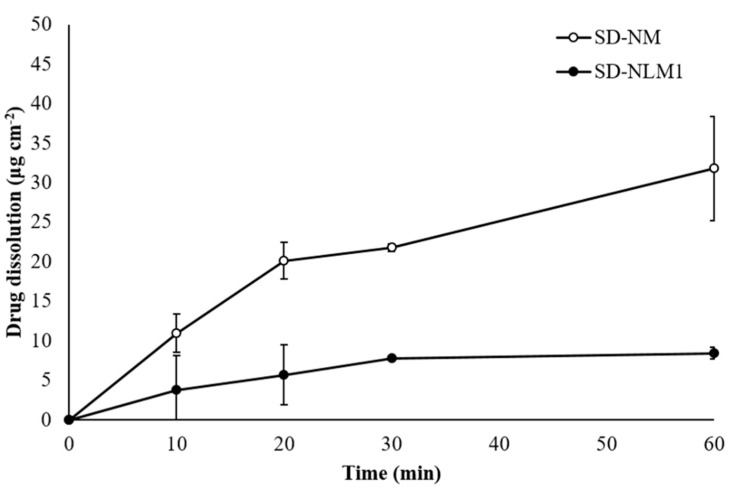
In vitro dissolution profile of spray dried NFM and mannitol (SD-NM) and spray dried microparticles containing NFM, lecithin, and mannitol (SD-NLM) (mean ± standard deviation, *n* = 3).

**Figure 5 pharmaceutics-13-01519-f005:**
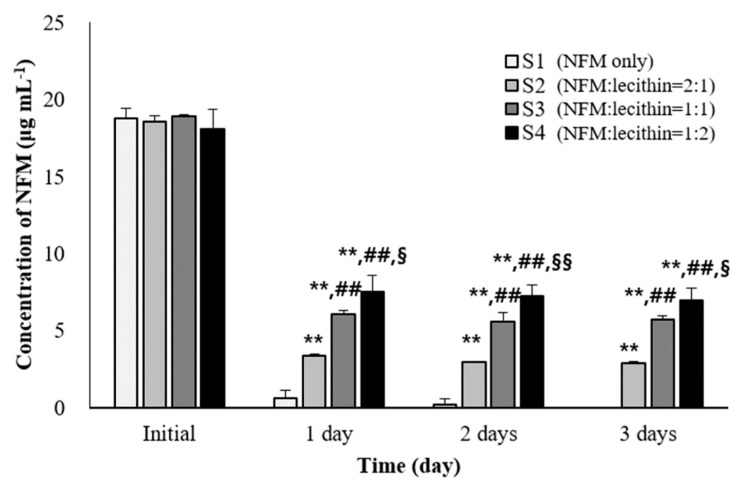
Ex vivo stability test of nafamostat mesylate (NFM) in bronchoalveolar lavage fluid (BALF) of Sprague–Dawley rats (mean ± standard deviation, *n* =3). ** ANOVA, *p*-value < 0.005 compared with S1 group; ## ANOVA, *p*-value < 0.005 compared with S2 group; §ANOVA, *p*-value < 0.05 compared with S3 group; §§ ANOVA, *p*-value < 0.005 compared with S3 group.

**Figure 6 pharmaceutics-13-01519-f006:**
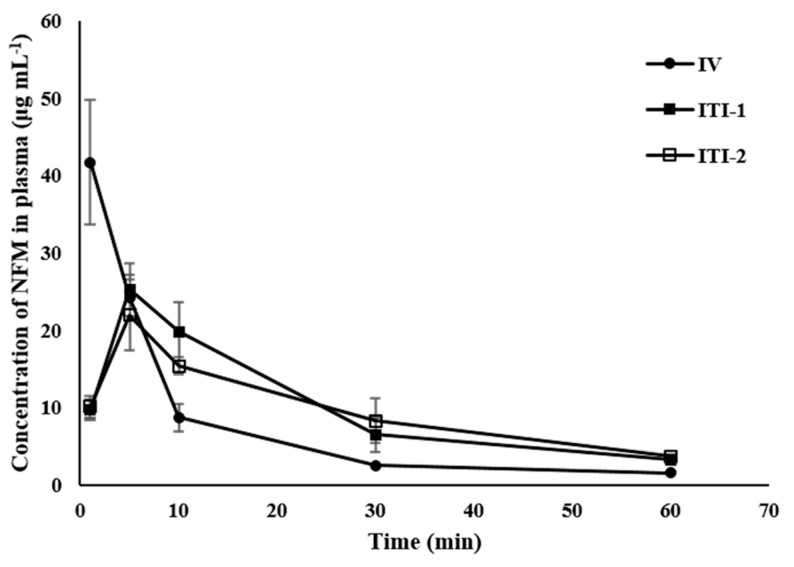
Mean plasma concentration versus time curves of nafamostat mesylate (NFM) after administration of 10 mg kg^−1^ NFM to Sprague–Dawley rats (mean ± standard error). ITI: intratracheal instillation; IV: intravenous; ITI-1: without lecithin; ITI-2: with lecithin.

**Figure 7 pharmaceutics-13-01519-f007:**
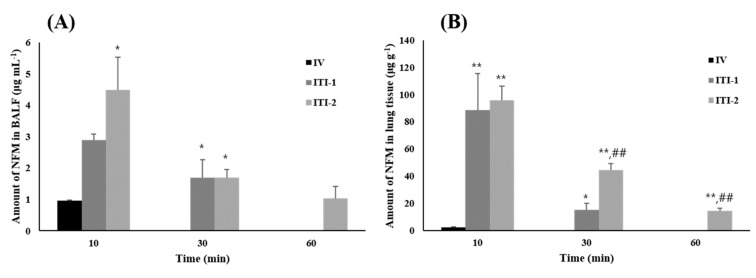
In vivo deposition amount of nafamostat mesylate (NFM) solution by administration route after administration of 10mg kg^−1^ NFM in Sprague–Dawley rats. **(A**) Amount of NFM in the bronchoalveolar lavage fluid (BALF) indicating amount of NFM in the lung lumen (mean ± standard error, *n* = 4), (**B**) Amount of NFM in lung tissue (mean ± standard error, *n* = 4). ITI: intratracheal instillation; IV: intravenous; ITI-1: without lecithin; ITI-2: with lecithin. * ANOVA, *p*-value < 0.05 compared with IV group; ** ANOVA, *p*-value < 0.005 compared with IV group; ## ANOVA, *p*-value < 0.005 compared with ITI-1 group.

**Figure 8 pharmaceutics-13-01519-f008:**
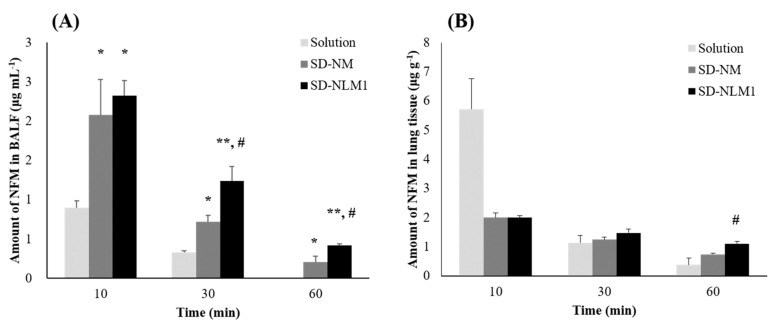
In vivo deposition amount of NFM microparticles (**A**) Amount of NFM in BALF after treatment with NFM microparticles (1mg/kg) indicating amount of NFM in the lung lumen (mean ± standard error, *n* = 4) (**B**) Amount of NFM in lung tissue after treatment with NFM microparticles (1mg/kg) (mean ± standard error, *n* = 4). Solution: intra-tracheal instillation; SD-NM: spray dried NFM and mannitol (1:100); SD-NLM1: spray dried NFM, lecithin and mannitol (1:1:100). * ANOVA, *p*-value < 0.05 compared with Solution group; ** ANOVA, *p*-value < 0.005 compared with Solution group; # ANOVA, *p*-value < 0.05 compared with SD-NM group.

**Table 1 pharmaceutics-13-01519-t001:** Yield and particle size distribution of nafamostat mesylate (NFM) microparticles (mean ± standard deviation, *n* = 3). SD-NM: spray dried NFM and mannitol (1:100); SD-NLM1: spray dried NFM, lecithin and mannitol (1:1:100); SD-NLM2: spray dried NFM, lecithin and mannitol (1:1:50); SD-NLM3: spray dried NFM, lecithin, and mannitol (1:1:10).

Results	SD-NM 1:1 (*w/w*)	SD-NLM1 1:1:100 (*w/w*)	SD-NLM2 1:1:50 (*w/w*)	SD-NLM3 1:1:10 (*w/w*)
Yield (%)	42.1	44.7	21.5	5.1
Dv10 (μm)	0.27 ± 0.002	1.36 ± 0.000	0.29 ± 0.001	1.35 ± 0.055
Dv50 (μm)	2.40 ± 0.021	2.60 ± 0.010	2.58 ± 0.015	10.27 ± 0.153
Dv90 (μm)	4.99 ± 0.090	4.87 ± 0.065	10.93 ± 0.252	103.47 ± 4.319
Span	1.97 ± 0.021	1.35 ± 0.022	4.13 ± 0.082	9.96 ± 0.246

**Table 2 pharmaceutics-13-01519-t002:** Aerosol performance characteristics of NFM microparticles including emitted dose (ED), fine particle fraction (FPF), mass median aerodynamic diameter (MMAD), and geometric standard deviation (GSD) (mean ± standard deviation, *n* = 3).

Results	SD-NM	SD-NLM1	SD-NLM2	SD-NLM3
ED (%)	93.7 ± 2.0 **^,##,§§^	98.6 ± 0.8	99.7 ± 0.4	100.0 ± 0.0
FPF (%)	75.1 ± 5.3**^,##,§§^	50.8 ± 3.5 **^,##^	24.4 ± 1.9	19.9 ± 0.6
Total deposited amount (%)	52.3 ± 5.4	74.6 ± 6.4 **	63.1 ± 4.9	59.0 ± 8.4 ^§^
MMAD (μm)	2.1 ± 0.6 **^,##,§^	3.5 ± 0.3 **^,##^	6.8 ± 0.8	6.4 ± 0.7
GSD	2.6 ± 0.3 ^##,§§^	3.0 ± 0.2 ^##,§§^	4.2 ± 0.3	3.9 ± 0.3

** ANOVA, *p*-value < 0.005 compared with SD-NLM3; ^##^ ANOVA, *p*-value < 0.005 compared with SD-NLM2; ^§^ ANOVA, *p*-value < 0.05 compared with SD-NLM1; ^§§^ ANOVA, *p*-value < 0.005 compared with SD-NLM1.

**Table 3 pharmaceutics-13-01519-t003:** Pharmacokinetic parameters of NFM solution (mean ± standard error, *n* = 4).

Results	IV	ITI-1	ITI-2
**AUC_(inf)_ (min·** **μg mL^−1^)**	436.1 ± 31.59	675.7 ± 50.83 *	730.0 ± 67.84 **
**C_max_** **(μg mL^−1^)**	43.7 ± 7.98	25.9 ± 4.23	22.9 ± 4.64 *
**T_max_** **(min)**	-	4.0 ± 1.00	10.3 ± 6.65
**T_½_** **(min)**	13.2 ± 0.52	26.7 ± 2.60 **	28.6 ± 3.51 **

* ANOVA, *p*-value < 0.05 compared with IV; ** ANOVA, *p*-value < 0.005 compared with IV.; AUC_inf_: area under the curve from dosing to time infinity; Cmax: maximum plasma concentration; ITI: intratracheal instillation; IV: intravenous; ITI-1: without lecithin; ITI-2: with lecithin; Tmax: time to maximum plasmatic concentration; T_½_ plasmatic half-life.

## Data Availability

Not applicable.

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
