# Peer review of "Co-Spray Dried Nafamostat Mesylate with Lecithin and Mannitol as Respirable Microparticles for Targeted Pulmonary Delivery: Pharmacokinetics and Lung Distribution in Rats"

_pharmaceutics, 2021, doi:10.3390/pharmaceutics13091519_

Round 1
Reviewer 1 Report
The manuscript by Kang et al. aims to develop an inhalable dry powder formulation of nafamostat mesylate with lecithin and mannitol using spray drying process. This study is well designed, and the topic covered is interesting to readers. The manuscript needs a major revision before accepting for the publication.
Abstract, Line 9: Correct the font case of lecithin.
Abstract, Line 11: The formulation produced with small particle size and excellent aerodynamic performance has lecithin or not? Please clarify.
Introduction, Paragraph 1, Line 6: Include the access date of reported statistics.
Section 2.1: Mention the purity details of NFM, and lecithin.
Section 2.2.1: Mention the % humidity in the glass vial containing silica gel.
Section 2.2.3: Justify the selection of 28.3 L/min air flow rate for the aerosolization studies. Also, I would recommend inclusion of HPLC method validation results.
Section 2.2.4: Justify the selection of distilled water as dissolution medium for the in vitro dissolution studies.
Section 2.2.5: Clarify the solvent used for the solution 1 preparation. Justify the selection of distilled water for BALF collection. Generally, PBS composed of 0.1 mM EDTA is used for the collection of BALF.
Section 2.2.6: This section needs to be rewritten as it is confusing in distinguishing the groups and the information.
Section 3: Results and discussion section needs to be rearranged as per the order mentioned in the methods section.
Section 3.1.: The stability of NFM in solution S3 and S4 are statistically same or different. Clarify.
Section 3.2: What is the effect of lecithin on the half-life of NFM?
Section 3.3.: Correct the composition of the SD particles in the first sentence.
Section 3.3: The formulation codes are confusing. Clarify the composition of formulation SD-NM.
Figure 4: Replace this figure with high magnification images.
Section 3.4: Rerun the DSC experiments to higher temperatures and check the effect of spray drying and components of the formulation on the melting behavior of drug. What is the melting point of drug and lecithin. Keep the technique names consistent XRD or PXRD. In methods, it has been mentioned as XRD and in results, it was mentioned as PXRD. FT-IR results needs to be reanalyzed and discussed appropriately as NFM and SD-NFM spectra shows a significant change in peak number and positions.
Figure 5: Correct the labelings.
Author Response
Reviewer 1
The manuscript by Kang et al. aims to develop an inhalable dry powder formulation of nafamostat mesylate with lecithin and mannitol using spray drying process. This study is well designed, and the topic covered is interesting to readers. The manuscript needs a major revision before accepting for the publication.
Abstract, Line 9: Correct the font case of lecithin.
:We have corrected what you mentioned.
Abstract, Line 11: The formulation produced with small particle size and excellent aerodynamic performance has lecithin or not? Please clarify.
:We modify the sentence to “The formulation prepared using an NFM:lecithin:mannitol ratio of 1:1:100 had a small particle size and excellent aerodynamic performance.”
Introduction, Paragraph 1, Line 6: Include the access date of reported statistics.
:We modify the sentence to “According to the World Health Organization (WHO 2020), approximately 70 million peo-ple have been infected with COVID-19 worldwide, and the death toll has reached 2.3 mil-lion in about 220 countries [1–3].”
Section 2.1: Mention the purity details of NFM, and lecithin.
:As recommended, we add purity on page 3, section 2.1 Materials.
Section 2.2.1: Mention the % humidity in the glass vial containing silica gel.
: As recommended, we add relative humidity on page 3, section 2.2.1.
Section 2.2.3: Justify the selection of 28.3 L/min air flow rate for the aerosolization studies. Also, I would recommend inclusion of HPLC method validation results.
:According to United States Pharmacopeia (USP) chapter 601, 28.3 LPM conditions are generally used to simulate minimum adult breathing with Andersen cascade impactor (ACI). We conducted a study at 28.3LPM to simulate a patient with reduced respiratory function due to corona virus infection.
:We add validation results of HPLC method on supplement.
Section 2.2.4: Justify the selection of distilled water as dissolution medium for the in vitro dissolution studies.
:First, we used distilled water (DW) to secure the sink condition of NFM. In addition, we proceeded on the basis of the references that performed the dissolution test using DW. We add reference on page 5, section 2.2.4.
Section 2.2.5: Clarify the solvent used for the solution 1 preparation. Justify the selection of distilled water for BALF collection. Generally, PBS composed of 0.1 mM EDTA is used for the collection of BALF.
:As recommended, we add DW. In addition, the reason for proceeding with BALF to DW is to secure sufficient capacity of solubility as mentioned above.
:EDTA was not added because there was a risk of affecting the HPLC analysis, and the protein was precipitated through the solvent precipitation method.
Section 2.2.6: This section needs to be rewritten as it is confusing in distinguishing the groups and the information.
:As recommended, we rewritten and re-revised the whole manuscript carefully and tried to avoid any grammar or syntax error. We attached the certificate of English editing for this manuscript.
Section 3: Results and discussion section needs to be rearranged as per the order mentioned in the methods section.
:As recommended, we reorganized the order of the method section and the result section in the same way.
Section 3.1.: The stability of NFM in solution S3 and S4 are statistically same or different. Clarify.
:They are statistically different. The statistical symbols have been rearranged in the graph (Figure 5 on page 12).
Section 3.2: What is the effect of lecithin on the half-life of NFM?
:As mentioned in Introduction (Page 2, last paragraph), it is thought that lecithin inhibited the degradation of NFM because it acts as an inhibitor of esterase.
Section 3.3.: Correct the composition of the SD particles in the first sentence.
:Following the comments, we modified the sentence (Section 2.2.1 on page 3).
Section 3.3: The formulation codes are confusing. Clarify the composition of formulation SD-NM.
:Following the comments, we clarified formulation codes (Section 2.2.4 on page 5)
Figure 4: Replace this figure with high magnification images.
:As recommended, we reconstructed the figure by replacing it with an image with high magnification (Figure 1 on page 7).
Section 3.4: Rerun the DSC experiments to higher temperatures and check the effect of spray drying and components of the formulation on the melting behavior of drug. What is the melting point of drug and lecithin. Keep the technique names consistent XRD or PXRD. In methods, it has been mentioned as XRD and in results, it was mentioned as PXRD. FT-IR results needs to be reanalyzed and discussed appropriately as NFM and SD-NFM spectra shows a significant change in peak number and positions.
:Following the comments, we made a graph again including the melting point of NFM (Figure 2 on page 8 and).
:We also added an interpretation of the DSC to the manuscript (on page 7).
:The terms were unified as XRD.
:Discussion on the difference between the FT-IR spectrum of NFM and SD-NFM has been added on page 8.
Figure 5: Correct the labelings.
:We modify the figure and labeling (Figure 2, on page 8)
Reviewer 2 Report
In this work, the Authors describe the production of a new dry formulation intended for pulmonary delivery based on nafamostat mesylate and lecithin as a synergic strategy for the treatment of Covid-19.
The article is generally well written and scientifically sound. Some minor comments are listed below:
- Introduction: “It may also increase the half-life of NFM when administered with ester group-rich substances such as lecithin, which could suppress the degradation of NFM by esterase in the respiratory lumen”. Please add a reference at the end of the period.
- Paragraph 2.2.3.: “Each formulation was aerosolized in an amount equivalent to 1 mg of NFM for each experiment….”.What does it mean? Did the Authors use a different amount of powder delivered in the ACI according to the amount of NFM in the different formulations? The different amount of delivered powder could impact the aerodynamic performance of the formulations, the powder loading should be kept constant. Please clarify.
- Paragraph 2.2.4.: Please state that SD-NM means spray-dried NFM+mannitol.
- Paragraph 2.2.6: Why was tween added in solution A?
Why did the author use a 0.25mg doses? Is there any reference?
- Paragraph 3.1.: The S4 solution seems to work better than S3. Why a NFM/Lecithin 1:2 (S4) was not used as ratio for DPI powder development?
- Paragraph 3.3.: “…microparticles containing NFM, mannitol, and mannitol…” Did the Authors mean lecithin? What did the Authors mean with "solid mass content"? Is it 2% w/v for all the solutions as declared in the method section? Please clarify this sentence. Please add a reference regarding the binding properties of lecithin.
- Paragraph 3.4.: Did the Authors spray-dried and characterized the mannitol alone? It should be added to the plots in fig. 5.
- Figure 5: Please change NAFA in NMF to be less confusing.
- Figure 6: Please add the percentage of powder that remained in the induction port and the capsule/device. Did the Authors perform the mass balance assay?
- Paragraph 3.7: The paragraph is more a PK study than an efficacy study.
Author Response
Reviewer 2
In this work, the Authors describe the production of a new dry formulation intended for pulmonary delivery based on nafamostat mesylate and lecithin as a synergic strategy for the treatment of Covid-19.
The article is generally well written and scientifically sound. Some minor comments are listed below:
Introduction: “It may also increase the half-life of NFM when administered with ester group-rich substances such as lecithin, which could suppress the degradation of NFM by esterase in the respiratory lumen”. Please add a reference at the end of the period.
:As recommended, we added reference (on page 2).
Paragraph 2.2.3.: “Each formulation was aerosolized in an amount equivalent to 1 mg of NFM for each experiment….”.What does it mean? Did the Authors use a different amount of powder delivered in the ACI according to the amount of NFM in the different formulations? The different amount of delivered powder could impact the aerodynamic performance of the formulations, the powder loading should be kept constant. Please clarify.
:We used a different amount of powder delivered in the ACI according to the amount of NFM in the different formulations. We considered and evaluated the amount of total powder to deliver the same amount of NFM as one of the formulation characteristics. The reason is that the same amount of API must be administered regardless of the formulation when finally administered to a patient as a product.
Paragraph 2.2.4.: Please state that SD-NM means spray-dried NFM+mannitol.
:Following the comments, we clarified on page 5
Paragraph 2.2.6: Why was tween added in solution A? Why did the author use a 0.25mg doses? Is there any reference?
:The Tween was added to evenly disperse the lecithin.
:The dose of 10 mg/kg (2.5mg/Rat) was established based on a reference administered by the IV route. We added the reference (on page 6).
Paragraph 3.1.: The S4 solution seems to work better than S3. Why a NFM/Lecithin 1:2 (S4) was not used as ratio for DPI powder development?
:Although the data was not shown, the flowability of the powder became very poor as the ratio of lecithin increased. This can also be confirmed in SD-NLM2 or SD-NLM3, which has a relatively high ratio of lecithin. In the SEM image (Figure 1, on page 7), it can be seen that they are formed cluster with each other.
Paragraph 3.3.: “…microparticles containing NFM, mannitol, and mannitol…” Did the Authors mean lecithin? What did the Authors mean with "solid mass content"? Is it 2% w/v for all the solutions as declared in the method section? Please clarify this sentence. Please add a reference regarding the binding properties of lecithin.
:It means lecithin. We modified the sentence (Section 3.1, on page 6).
:The solid mass content mentioned in that sentence means the ratio of NFM and mannitol. The sentence was changed to "As the proportion of liquid lecithin increased, the yield decreased" (Section 3.1 on page 6).
:We modified the sentence “Because of the viscosity of lecithin, particles form clusters and exist in a large particle.” (Section 3.1 on page 6).
Paragraph 3.4.: Did the Authors spray-dried and characterized the mannitol alone? It should be added to the plots in fig. 5.
:As recommended, we added spray dried mannitol data (Figure 2).
Figure 5: Please change NAFA in NMF to be less confusing.
:We modified labeling (Figure 2).
Figure 6: Please add the percentage of powder that remained in the induction port and the capsule/device. Did the Authors perform the mass balance assay?
:Unfortunately, Induction port and device were not analyzed separately. However, the amount remaining in the capsule was analyzed, and the emit dose (ED) value was calculated from the filling amount minus capsule residual amount. Therefore, other than the amount calculated from the analysis, it is assumed that the remaining particles are in the induction port or device (Table 2).
Paragraph 3.7: The paragraph is more a PK study than an efficacy study.
:As recommended, we modified (Section 3.7 on page 13).
Round 2
Reviewer 1 Report
I would recommend accepting this manuscript in the present form.